# DNA Methylation-Based Estimates of Circulating Leukocyte Composition for Predicting Colorectal Cancer Survival: A Prospective Cohort Study

**DOI:** 10.3390/cancers13122948

**Published:** 2021-06-12

**Authors:** Xīn Gào, Yan Zhang, Xiangwei Li, Lina Jansen, Elizabeth Alwers, Melanie Bewerunge-Hudler, Matthias Schick, Jenny Chang-Claude, Michael Hoffmeister, Hermann Brenner

**Affiliations:** 1Division of Clinical Epidemiology and Aging Research, German Cancer Research Center (DKFZ), Im Neuenheimer Feld 581, 69120 Heidelberg, Germany; gzhyanzi@hotmail.com (Y.Z.); xiangwei.li@dkfz.de (X.L.); l.jansen@dkfz.de (L.J.); elizabeth.alwers@dkfz.de (E.A.); m.hoffmeister@dkfz.de (M.H.); h.brenner@dkfz.de (H.B.); 2German Cancer Consortium, German Cancer Research Center (DKFZ), Im Neuenheimer Feld 280, 69120 Heidelberg, Germany; 3Medical Faculty, University of Heidelberg, 69120 Heidelberg, Germany; 4Genomics and Proteomics Core Facility, German Cancer Research Center (DKFZ), Im Neuenheimer Feld 580, 69120 Heidelberg, Germany; m.hudler@dkfz.de (M.B.-H.); m.schick@dkfz.de (M.S.); 5Division of Cancer Epidemiology, German Cancer Research Center (DKFZ), Im Neuenheimer Feld 581, 69120 Heidelberg, Germany; j.chang-claude@dkfz.de; 6Division of Preventive Oncology, German Cancer Research Center (DKFZ) and National Center for Tumor Diseases (NCT), Im Neuenheimer Feld 460, 69120 Heidelberg, Germany

**Keywords:** DNA methylation, circulating leukocytes, colorectal cancer, prognosis

## Abstract

**Simple Summary:**

Inflammation is involved in the evolution of cancer. Leukocytes, of which the proportion can be estimated using epigenome-wide methylation data, may serve as a prognostic marker in colorectal cancer (CRC). Our aim was to investigate whether DNA methylation-based estimates of circulating leukocytes is associated with all-cause and disease-specific mortality in a prospective CRC patients’ cohort. Significant associations with CRC prognosis were observed for CD4+ T cells, CD8+ T cells, B cells, NK cells, and lymphocytes, independent of age, sex, tumor stage, tumor subsite, and therapy. CD4+ T cells outperformed other leukocytes and provided added predictive value in comparison to age, sex, and tumor stage. Although cell counting is commonly used in clinical practice, DNA methylation-estimated cell proportions could be a promising tool in understanding the role of leukocytes as CRC prognostic biomarkers when using stored blood samples.

**Abstract:**

Leukocytes are involved in the progression of colorectal cancer (CRC). The proportion of six major leukocyte subtypes can be estimated using epigenome-wide DNA methylation (DNAm) data from stored blood samples. Whether the composition of circulating leukocytes can be used as a prognostic factor is unclear. DNAm-based leukocyte proportions were obtained from a prospective cohort of 2206 CRC patients. Multivariate Cox regression models and survival curves were applied to assess associations between leukocyte composition and survival outcomes. A higher proportion of lymphocytes, including CD4+ T cells, CD8+ T cells, B cells, and NK cells, was associated with better survival, while a higher proportion of neutrophils was associated with poorer survival. CD4+ T cells outperformed other leukocytes in estimating the patients’ prognosis. Comparing the highest quantile to the lowest quantile of CD4+ T cells, hazard ratios (95% confidence intervals) of all-cause and CRC-specific mortality were 0.59 (0.48, 0.72) and 0.59 (0.45, 0.77), respectively. Furthermore, the association of CD4+ T cells and prognosis was stronger among patients with early or intermediate CRC or patients with colon cancer. In conclusion, the composition of circulating leukocytes estimated from DNAm, particularly the proportions of CD4+ T cells, could be used as promising independent predictors of CRC survival.

## 1. Introduction

Colorectal cancer (CRC) represents the second leading cause of cancer-related deaths worldwide [1]. Although recent improvements have been made in screening strategies and treatments for CRC, the prognosis of advanced CRC is still poor [2,3]. Moreover, prognostic information provided by the anatomy-based tumor-node-metastasis staging system is incomplete. Survival rates can be significantly different among patients within the same stage [4]. Therefore, novel prognostic markers are needed to improve patient management for tertiary prevention and to elicit a better understanding of the disease processes. We previously observed that higher levels of DNA methylation (DNAm)-based mortality risk score, AgeAccelPheno and AgeAccelGrim were statistically significantly associated with poorer CRC prognosis. However, the associations were weaker for CRC-specific survival than for overall survival. Studies are needed to identify prognostic biomarkers that are more specific to CRC [5].

Immune inflammatory cells are involved in CRC progression in conflicting ways: both tumor-suppressing and tumor-promoting leukocytes are involved [6,7]. Intratumoral lymphocytes can induce tumor cell death by secreting tumor necrosis factor-α and interferon-γ [8]. On the other hand, a higher infiltration of neutrophils facilitates the pathological angiogenesis that is related to tumor growth [9]. Epidemiological studies have shown that a higher neutrophil to lymphocyte ratio in peripheral blood is associated with a worse CRC survival [10,11,12,13,14]. These findings suggest that the leukocyte composition may represent the underlying immuno-biology to CRC progression and could serve as a promising prognostic marker for CRC [15]. However, usually, freshly drawn venous blood and strict cell processing are critical for quantifying peripheral blood leukocyte composition using traditional techniques. Therefore, leukocyte counts are often missing in large-scale clinical and epidemiological studies.

Cell lineages can be distinguished by differentially methylated DNA regions. Methods have been developed to infer the leukocyte composition using DNA methylation (DNAm) signatures that are chemically stable [16,17,18]. Salas and colleagues identified a library of 450 CpGs for deconvoluting CD4+ T cells, CD8+ T cells, B cells, natural killer (NK) cells, monocytes and neutrophil fraction, which were specific to DNAm signatures assayed using Illumina Infinium HumanMethylationEPIC BeadChip Kit (EPIC microarray, Illumina, Inc., San Diego, CA, USA) [18]. The method showed good accuracy of cell composition estimates, and the average coefficient of determination was 99.2% between the artificial predefined cell proportions of blood mixtures and the estimations [18]. It was further validated in actual blood samples and publicly available datasets [18].

In this work, the leukocyte composition was estimated using Salas’ method based on EPIC microarray data in a large cohort of CRC patients. We then investigated the association of leukocyte composition with all-cause mortality and CRC-specific mortality of CRC patients and their capability for predicting survival.

## 2. Materials and Methods

### 2.1. Study Design and Population

The current study is based on the prospective follow-up of CRC patients of an ongoing population-based case-control study on CRC, the German Darmkrebs: Chancen der Verhütung durch Screening (DACHS) Study. Details of the DACHS study design have been described elsewhere [19,20,21,22]. Briefly, we recruited patients with a first diagnosis of CRC (ICD 10 codes: C18–C20) in 22 clinics that provide CRC surgery in the Rhine-Neckar region of Germany. In the current analysis, we included 2206 CRC patients whose information on follow-up with respect to survival outcomes and DNAm array data from blood samples taken at baseline.

### 2.2. Data Collection

CRC patients were diagnosed between 2003 and 2010 and were invited by their treating physicians during the first hospital stay due to CRC and notified to the study center at the German Cancer Research Center, after receiving informed consent. Trained interviewers collected patients’ sociodemographic, medical and lifestyle information using a standardized questionnaire at the earliest possible convenience, either during hospital stay or shortly thereafter at patients’ homes. Moreover, data on tumor characteristics and treatment were extracted from medical records. Information on newly diagnosed diseases and recurrences was provided by physicians at 3- and 5-year follow-up. Data on date and cause of death were obtained from the local population registers and public health authorities. The study was approved by the Ethics Committees of the Medical Faculty of the University of Heidelberg (ID: 310/2001) and the Medical Chambers of Baden-Württemberg (ID: M-198-02) and Rhineland-Palatinate (ID: 837.419.02 (3637)).

### 2.3. DNA Methylation and Leukocyte Composition Estimation

Blood samples were collected after the interview and were shipped to the study center and stored at −80 °C until whole blood DNA extraction. Genome-wide DNAm signatures were assayed using the EPIC microarray (Illumina, Inc., San Diego, CA, USA) according to the manufacturer’s protocol at the Genomics and Proteomics Core Facility of the German Cancer Research Center. Preprocessing and normalization of DNAm data was conducted following the pipeline proposed by Lehne and colleagues [23]. Probes with detection *p* value > 0.01, with missing value > 10%, probes targeting the sex chromosomes, cross-reactive probes and polymorphic CpGs were excluded, leaving 787,231 CpGs for analyses. The proportions of CD4+ T cells, CD8+ T cells, B cells, NK cells, monocytes and neutrophils were estimated using a deconvolution method, which was implemented with the FlowSorted.Blood.EPIC package of R [24].

### 2.4. Statistical Methods

The proportion of categorical covariates was calculated to describe the characteristics of the study population. The distribution of baseline characteristics of patients at different stages were compared using Chi square test. Wilcoxon−Mann−Whitney test was performed to compare medians of leukocyte composition across categorical variables. Jonckheere−Terpstra test was used to determine the significance of the trend of ordinal variables. Correlations between leukocyte components were assessed using Pearson correlation coefficients and scatter plots.

Associations between leukocyte composition and survival were assessed using Cox regression, from which hazard ratios (HRs) and 95% confidence intervals (95% CIs) were derived. We ran a “clinical model” adjusted for covariates that can be easily obtained in clinical settings, including age, sex, stage, CRC subsite, batch, and timing of blood sampling. A “comprehensive model” additionally adjusted for body mass index (BMI, kg/m^2^), smoking status (never, former and current smokers), and Charlson comorbidity index (CCI) score was applied to evaluate potential confounding by these covariates. The characteristics of all covariates except for the timing of blood sampling are shown in Table 1. Complete case analysis was applied to address missing data (<0.5%) when adjusting for these variables. The timing of blood sampling was grouped into four categories including: (1) prior to surgery or within 1 month after surgery (*n* = 1006); (2) more than 1 month after surgery, but not receiving chemotherapy or radiotherapy (*n* = 569); (3) during chemo-or radiotherapy, or within 5 months after the first treatment (*n* = 278); (4) more than 5 months after chemo-or radiotherapy (*n* = 353) (Appendix A). The proportional hazards assumption was tested using the Schoenfeld Residuals. Furthermore, we accounted for delayed entry time by incorporating the time elapsed between diagnosis and the starting of patients’ enrolment in the standard Cox models.

Adjusted survival curves were plotted to assess whether the association differed depending on tumor stage, tumor subsite and the timing of blood sampling. The difference between survival curves across quartiles of leukocyte composition was evaluated using the G-rho family of tests. Furthermore, predictive accuracy and discriminating ability of leukocyte composition were evaluated using Harrell’s concordance statistics (C-statistics) and were compared with age, sex and stage. Cubic spline models were also performed to assess the dose−response relationship between CD4+ T cell proportion and CRC mortality in various therapy groups.

Correlation matrix, survival curves and cubic spline curves were produced using the R 3.6.0 with the packages corrplot, survminer and survival, respectively [25]. Hazard ratios and Harrell’s C-statistics were derived using the PROC PHREG procedure in SAS version 9.4 (SAS Institute, Cary, NC, USA). Statistical significance was defined by *p* < 0.05 in two-sided testing.

## 3. Results

### 3.1. Characteristics of the Study Population

Table 1 summarizes the demographic and clinical characteristics of the study population at baseline. Among the 2206 CRC patients, there were more men (58.8%) than women (41.2%). More than two-thirds of the study population (67.5%) were older than 65 years, 62.0% of patients were overweight or obese, 15.9% of patients were current smoker at baseline, and more than 40% had a relevant comorbidity (CCI > 0). Of the patients, 18.2%, 34.6%, 33.1% and 14.1% were diagnosed with stage I, II, III and IV cancer, respectively. Colon cancer and rectal cancer accounted for 69.6% and 30.4% of CRC patients, respectively. The distribution of age, BMI, smoking status, CCI and tumor sub-site were statistically significantly different across the tumor stages (Appendix A). A lower lymphocyte proportion and a higher neutrophil proportion were observed in males, older patients, and patients diagnosed at advanced stages, with rectal cancer and with higher CCI (Appendix A). Moreover, CD4+ T cells, CD8+ T cells, B cells, NK cells, and monocytes were statistically positively correlated with each other, whereas neutrophils were statistically inversely correlated with other leukocyte subtypes (Appendix A).

### 3.2. Association of Leukocyte Composition with CRC Prognosis

Figure 1 shows that proportions of all leukocyte subtypes except monocytes were statistically significantly associated with all-cause mortality of CRC patients. HRs (95% CIs) for the association of all-cause mortality with highest (vs. lowest) quartiles of the proportions were: CD4+ T cells, 0.59 (0.48, 0.72); CD8+ T cells, 0.82 (0.67, 0.99); B cells, 0.68 (0.56, 0.83); NK cells, 0.88 (0.72, 1.07); monocytes, 0.98 (0.81, 1.18); and neutrophils, 1.55 (1.25, 1.91), respectively. However, only CD4+ T cell, B cell and neutrophil proportions were statistically significantly associated with CRC-specific mortality (Figure 2). HRs (95% CIs) for the comparison of the highest with the lowest quartile of the proportions were: CD4+ T cells, 0.59 (0.45, 0.77); CD8+ T cells, 0.88 (0.69, 1.14); B cells, 0.70 (0.54, 0.90); NK cells, 1.03 (0.79, 1.36); monocytes, 1.06 (0.83, 1.35); and neutrophils, 1.32 (1.00, 1.75), respectively. Furthermore, sensitivity analyses showed that some of the associations between leukocyte proportions and CRC prognosis were attenuated after additional adjustment for BMI, smoking status and CCI. However, statistically significant associations with CRC prognosis remained for CD4+ T cells, B cells and neutrophils (Appendix A).

### 3.3. Subgroup Analyses

Figure 3 presents the overall and CRC-specific survival curves across quartiles of the proportion of CD4+ T cells by tumor stage. We observed statistically significant associations between a higher proportion of CD4+ T cells and improved overall survival in the early and intermediate stages (I–III) but not in the advanced stage (IV). Moreover, the proportion of CD4+ T cells was statistically significantly positively associated with CRC-specific survival in patients with stage III cancers.

Figure 4 shows the tumor subsite-specific association of the proportion of CD4+ T cells with overall and CRC-specific survival. Among patients with colon cancer, a higher proportion of CD4+ T cells was statistically significantly associated with better survival, and the association was weaker for CRC-specific survival than for overall survival. However, the association between the proportion of CD4+ T cells and both outcomes was not statistically significant among patients with rectal cancer.

Additionally, patients in the second-highest quartile of CD4+ T cell proportion had the poorest survival among the subgroup whose blood was sampled during or within 5 months after initiation of chemo- or radiotherapy treatment, which is different from the other subgroups (Appendix A). Moreover, as shown in the dose−response curves, a “U-shaped” dose−response relationship was observed among patients who received surgery, chemotherapy and radiotherapy (or only received surgery and radiotherapy) (Appendix A).

### 3.4. Predictive Utility of CD4+ T Cell Proportion and All Leukocyte Proportions Combined

Table 2 presents the discriminative ability of various combinations of predictors, including age, sex, tumor stage, and leukocyte compositions. The prediction of CRC prognosis was modestly improved after adding the proportion of CD4+ T cells into the model, overall and by tumor stage. In all stages, as well as stages I–II and stage III, simultaneous consideration of all leukocyte subtypes combined slightly outperformed exclusive consideration of CD4+ T cell proportions for prediction of CRC prognosis. However, in stage IV, the models including CD4+ T cell proportions only, had an even better performance than the models including all leukocyte subtypes.

## 4. Discussion

This analysis is the first prospective study to investigate the association of DNAm-based estimates of leukocyte composition with CRC prognosis. Patients with higher estimated proportions of CD4+ T cells, CD8+ T cells and B cells had lower all-cause and CRC-specific mortality, and patients with a higher proportion of neutrophils had higher all-cause and CRC-specific mortality. Among all leukocyte subtypes, CD4+ T cells presented the strongest association with CRC prognosis. A one-SD increment of the proportion of CD4+ T cells (7%) was associated with 17% and 18% lower all-cause and CRC-specific mortality, respectively. Subgroup analyses showed that the association of CD4+ T cells with CRC prognosis was statistically significant in stage I–III and colon cancer patients.

CD4+ T cells, including T helper (Th) cells and regulatory T (Treg) cells, function to instigate and shape adaptive immune responses. Th cells can target tumor cells in various ways, either directly by eliminating tumor cells through cytolytic mechanisms or indirectly by activating and directing innate immune cells, B cells and CD8+ T cells [26]. Studies have shown that intratumoral CD4+ T cell infiltration can suppress tumor progression [27,28]. On the other hand, Treg cells, which are essential to maintain self-tolerance and immune cell homeostasis, are involved in the tumor progression by establishing an immunosuppressive tumor microenvironment [29]. Suppression of tumor-specific CD4+ T cells by Treg was associated with a worse CRC prognosis in a cohort of 62 patients [30]. CD8+ T cells, also known as cytotoxic T lymphocytes, can infiltrate into tumor tissue and kill malignant cells by releasing cytotoxic granules or by recognizing Fas ligand on the surface of target cells. A higher density of intratumoral CD8+ T cells was associated with improved CRC prognosis, which is in line with our findings [31,32].

B lymphocytes are involved in cell-mediated adaptive immunity and can inhibit tumor progression by secreting tumor-reactive antibodies and priming T cells [33]. Various studies reported that the tumor-infiltrating B cell density was positively associated with CRC prognosis [34,35]. However, regulatory B cells can suppress Th1 and CD8+ T cell responses and subsequently promote tumor progression, similar to Treg cells [33]. Nevertheless, evidence on the association of circulating CD4+ and CD8+ T cells and B cells with CRC prognosis is scarce. Further studies are needed to confirm our findings.

NK cells can kill tumor cells without any priming or prior activation. Tang and colleagues observed in 447 CRC patients that patients with an increased percentage of peripheral blood NKs had a higher 3-year survival rate [36]. It was consistent with our results that a marginal inverse association between NK proportion and all-cause mortality was observed among CRC patients whose circulating leukocyte composition was not affected by the therapies. Monocytes are a major innate immune component of the mononuclear phagocyte system. A meta-analysis of 16 prospective studies comprising 3826 CRC patients showed that increased absolute blood monocyte counts were significantly associated with worse overall survival [37]. However, no significant association between the proportion of blood monocytes and CRC prognosis was observed in our study. The difference between the results of the two studies could be explained by the different measures of peripheral blood monocyte (cell count vs. cell proportion).

Neutrophils are the most abundant leukocytes in blood and an essential part of the innate immune system. In line with our findings, a higher neutrophil to lymphocyte ratio in peripheral blood is associated with higher mortality among CRC patients, overall or by tumor stage, which suggests that lymphocytes and neutrophils may play opposite roles in CRC progression [10,11,12,13,14]. Evidence also shows that neutrophils may play both pro- and antitumor roles, depending on the heterogeneous subsets [38,39]. The association between neutrophil levels and CRC prognosis is controversial. Rao et al. observed that elevated intratumoral neutrophils in CRC were significantly associated with malignant phenotype and adverse prognosis in 229 patients [40]. However, Wikberg et al. found in a study of 448 patients that low infiltration of neutrophils in the tumor front was indicative of a worse prognosis [41]. Thus, research with larger CRC patient cohorts is needed to investigate the relationship of neutrophils with survival.

DNAm-base markers have shown prognostic value for CRC patients in previous studies [5,42]. Our study further revealed that blood DNAm-based leukocyte composition, especially the proportion of CD4+ T cells, could be used as an independent marker to enhance clinical judgment of prognosis in patients with early or intermediate stage CRC. Leukocyte composition can also be used to explore potential mechanisms underlying immune responses to tumor growth. The positive association of each subtype of circulating lymphocytes with CRC overall survival suggests that lymphocytes, at the interface of innate and adaptive immunity, suppress CRC progression. Our data suggest that specific lymphocytes and neutrophils may be potentially utilized as pathogenic effectors and therapeutic targets.

CD4+ T cells were the strongest predictor for CRC prognosis among leukocyte subtypes in our study. However, the associations between CD4+ T cell proportion and CRC survival outcomes were weak and statistically nonsignificant among patients with advanced CRC. A possible explanation is that the survival for advanced CRC is generally extremely poor and the case numbers in this group were the smallest, which limited the statistical power to detect possible associations. Among early- and intermediate-stage patients, the predictive performance of circulating CD4+ cell proportion was better for all-cause mortality than for CRC-specific mortality, even after controlling for comorbidity score. It suggested that circulating CD4+ cell may not serve as a specific predictor for CRC prognosis. In subsite specific analyses, the association of a higher proportion of CD4+ T cells with better survival was statistically significant among colon cancer patients but not rectal cancer patients, of whom the sample size was smaller.

This study has several strengths including the prospective study design, large sample size, long-term follow-up, the well-recorded causes of death, and detailed information on the study population. Moreover, the composition of the leukocyte population was estimated using DNAm array data that is stable and can be obtained from stored whole blood samples.

There are also limitations. First, blood samples were obtained at various points of time shortly before or after surgery and potentially other therapy. However, leukocyte composition can be affected by surgery, chemo- and radiotherapy administration [43]. The percentage of neutrophils was higher, and the percentage of lymphocytes was lower shortly after surgery and chemotherapy or radiotherapy (Appendix A). We therefore adjusted for the timing of blood sampling relative to treatment in all Cox models. Survival curves stratified by the timing were performed. Significant and consistent associations of CD4+ T cell proportions with CRC prognosis were observed among all groups of patients except those whose blood sample was taken during, or within 5 months of the initiation of, chemo- or radiotherapy. These patterns suggest that one should avoid collecting blood samples within the course of chemo- or radiotherapy when using leukocyte estimates to predict CRC prognosis in research and clinical practice. Second, the absolute leukocyte counting information is missing in our study. Even though the DNAm-based estimation of leukocyte composition has been validated in various studies, comparison of the DNAm-based method with the traditional cell counting method is still needed in further studies. However, DNA methylation estimated cell proportions could help scientists to further understand the role of cell counts as prognostic biomarkers in CRC by using archived blood samples. Third, CRC is recognized as a heterogeneous disease, and its behavior differs between molecular subtypes [44], information on which was only available for a proportion of our patients and did not allow sufficiently powered analyses in our study. Further studies with comprehensive information on CRC molecular subtypes are needed to investigate the association across molecular subtypes. Lastly, the current high cost of DNAm microarray makes this technique difficult to be promoted in clinical settings. However, this study provides a potential tool to improve the prediction of CRC prognosis and to assist the physician’s decision-making. It also provides a foundation for further study of inspiration. The DNAm-based markers may gain wider use if a low-price DNAm microarray technique is developed and commercialized.

## 5. Conclusions

Our findings suggest that blood DNAm-based estimates of leukocyte distribution, in particular the proportion of CD4+ T cells, have the potential to improve the accuracy of prognostic judgment for patients with early and intermediate CRC. However, validation by other studies, in particular studies with information on molecular subtypes of CRC and leukocyte count, is warranted to corroborate and extend the clinical importance of this observation.

## Figures and Tables

**Figure 1 cancers-13-02948-f001:**
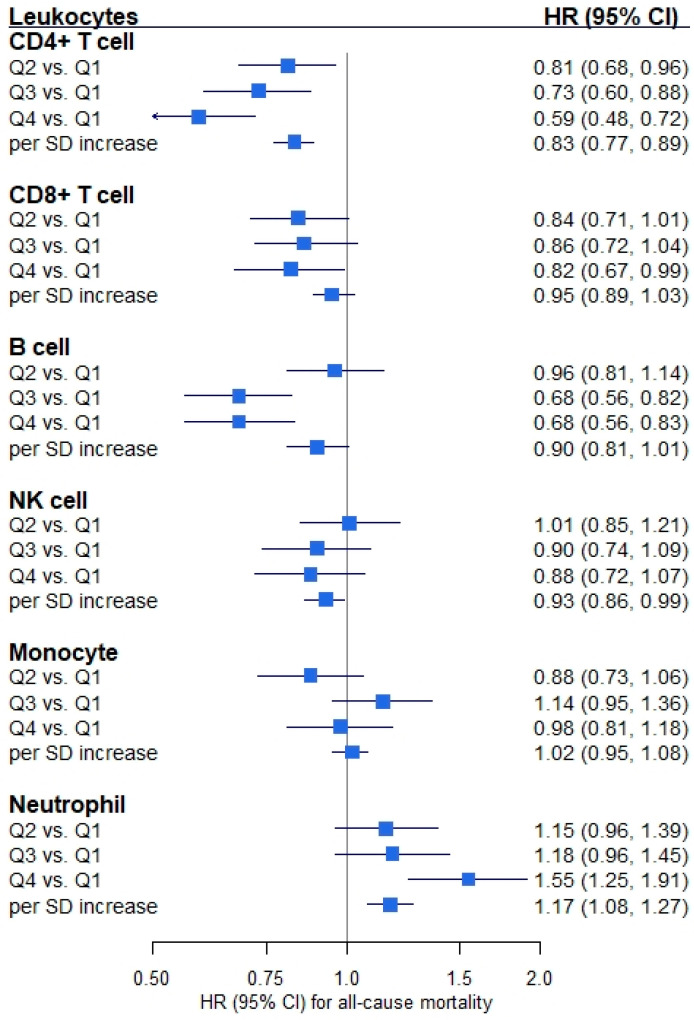
Forest plot of the associations between leukocyte composition and all-cause mortality. Note: Hazard ratio (HR) and 95% confidence interval (CI) were derived from Cox regression model adjusting for age, sex, tumor stage, tumor subsite, timing of blood sampling and batch (*n*_patients_ = 2194, *n*_deaths_ = 1071). An HR of one means that there is no difference in survival between the two groups. Abbreviation: Q1–Q4, quartile 1 (lowest)–quartile 4 (highest); SD, standard deviation.

**Figure 2 cancers-13-02948-f002:**
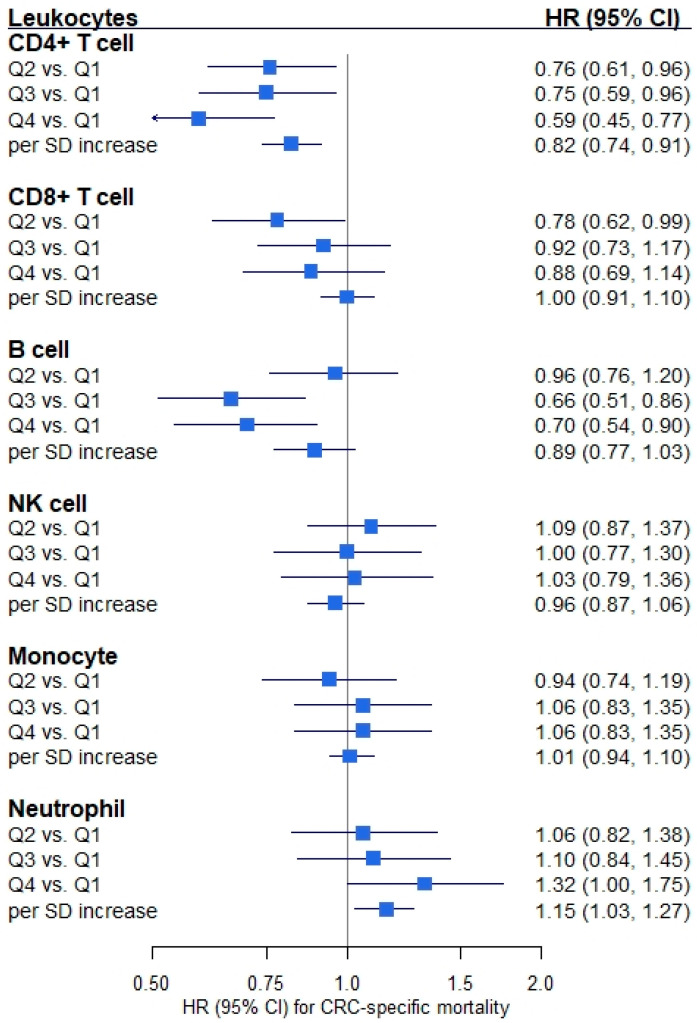
Forest plot of the associations between leukocyte composition and colorectal cancer (CRC)-specific mortality. Note: Hazard ratio (HR) and 95% confidence interval (CI) were derived from Cox regression model adjusting for age, sex, tumor stage, tumor subsite, timing of blood sampling and batch (*n*_patients_ = 2179, *n*_deaths_ = 593). An HR of one means that there is no difference in survival between the two groups. Abbreviation: Q1–Q4, quartile 1 (lowest)–quartile 4 (highest); SD, standard deviation.

**Figure 3 cancers-13-02948-f003:**
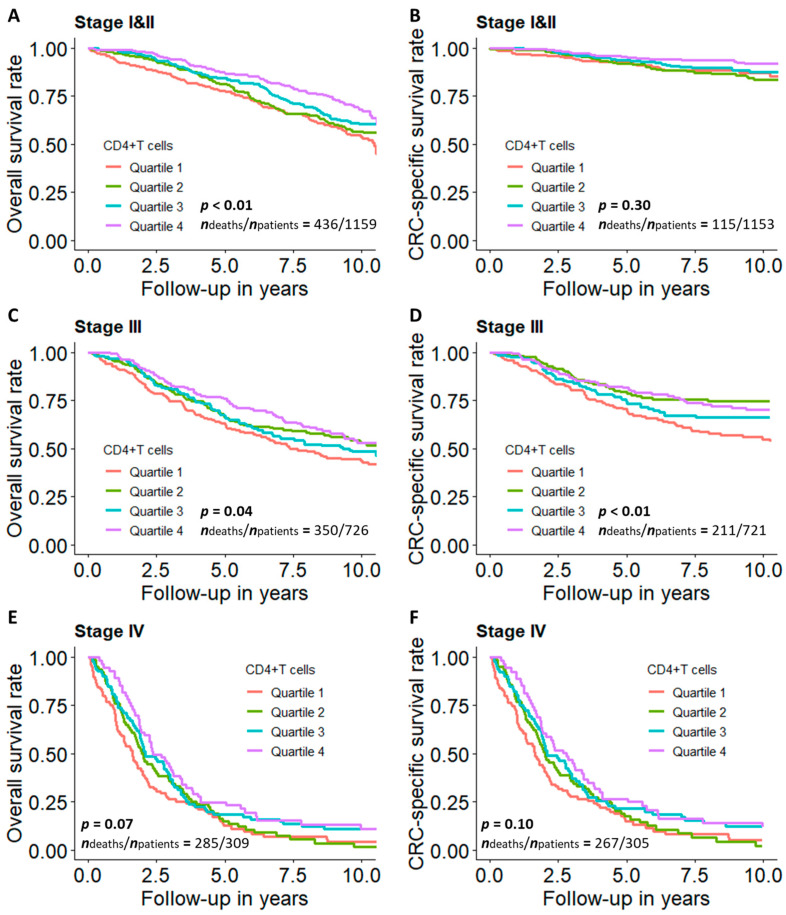
Survival curves for CRC patients across quartiles of CD4+ T cell proportions by tumor stage. (**A**) Overall and (**B**) CRC-specific survival curves among stage I and II patients; (**C**) overall and (**D**) CRC-specific survival curves among stage III patients; (**E**) overall and (**F**) CRC-specific survival curves among stage IV patients. Note: Survival curves were adjusted for age, sex, tumor subsite, timing of blood sampling and batch. *p* values were derived from G-rho family of tests. Quartile 1: lowest quartile; Quartile 4: highest quartile.

**Figure 4 cancers-13-02948-f004:**
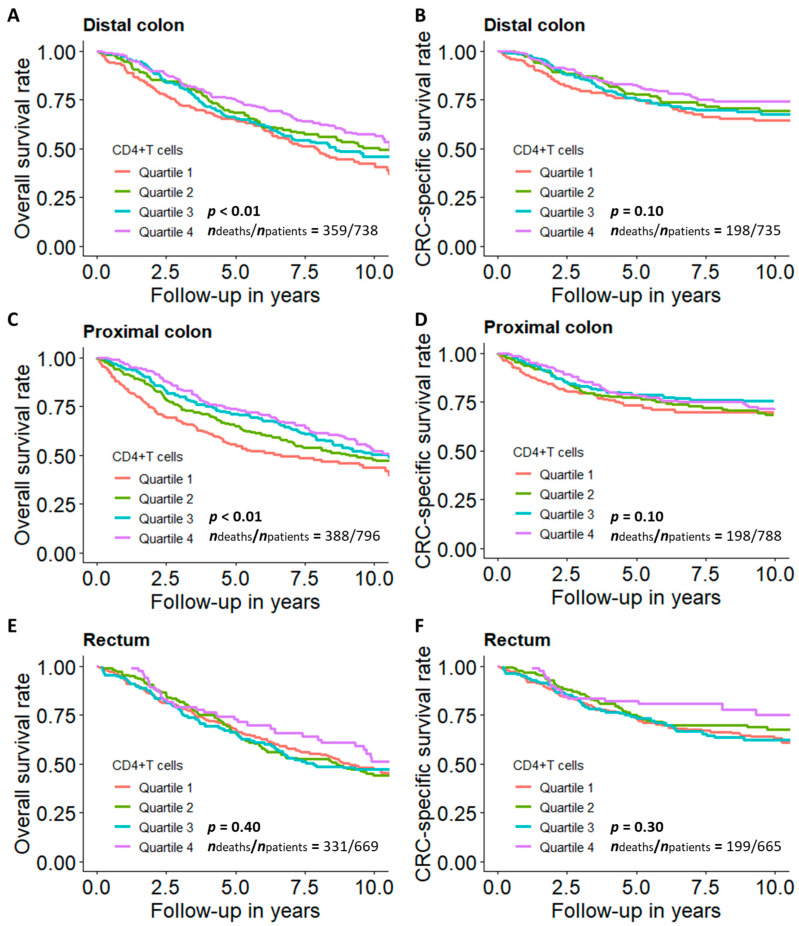
Survival curves for CRC patients across quartiles of CD4+ T cell proportions by tumor subsite. (**A**) Overall and (**B**) CRC-specific survival curve among patients with distal colon cancer; (**C**) overall and (**D**) CRC-specific survival curve among patients with proximal colon cancer; (**E**) overall and (**F**) CRC-specific survival curve among patients with rectal cancer. Note: Survival curves were adjusted for age, sex, tumor stage, timing of blood sampling and batch. *p* values were derived from G-rho family of tests. Quartile 1: lowest quartile; Quartile 4: highest quartile.

**Table 1 cancers-13-02948-t001:** Baseline characteristics of the patient cohort.

Characteristics	*n* (%) *
Sex	
Women	910 (41.2)
Men	1296 (58.8)
Age at diagnosis	
[33, 65) years	717 (32.5)
[65, 75) years	776 (35.2)
[75, 96) years	713 (32.3)
Tumor stage	
I	400 (18.2)
II	759 (34.6)
III	726 (33.1)
IV	309 (14.1)
Tumor subsite	
Distal colon †	738 (33.5)
Proximal colon ‡	796 (36.1)
Rectum	669 (30.4)
Body mass index at diagnosis	
<25 kg/m^2^	834 (38.0)
25–30 kg/m^2^	932 (42.4)
>30 kg/m^2^	430 (19.6)
Smoking status	
Never	907 (41.1)
Former	947 (43.0)
Current	350 (15.9)
Charlson comorbidity index	
0 (no comorbidity)	1281 (58.1)
1 (mild comorbidity)	479 (21.7)
2+ (moderate comorbidity)	446 (20.2)

Note: * 12 missing values for tumor stage, 10 missing values for body mass index, 2 missing values for smoking status and 3 missing values for tumor subsite. Complete case analysis was applied when adjusting for the variables with missing values. † The distal colon includes the descending colon and the sigmoid colon. ‡ The proximal colon includes the cecum, the ascending colon, the right flexure, and the transverse colon and the left flexure.

**Table 2 cancers-13-02948-t002:** Harrell’s C-statistics (95% confidence interval) for all-cause mortality and colorectal cancer (CRC)-specific mortality prediction.

	Combination of Predictors	All-Cause Mortality	CRC-Specific Mortality
All stages	Age + sex + stage	0.739 (0.723, 0.754)	0.809 (0.792, 0.825)
	Age + sex + stage + CD4(+)T cell	0.743 (0.728, 0.758)	0.813 (0.797, 0.829)
	Age + sex + stage + all leukocyte subtypes	0.744 (0.729, 0.759)	0.814 (0.798, 0.830)
Stage I and II	Age + sex	0.693 (0.667, 0.718)	0.612 (0.559, 0.666)
	Age + sex + CD4(+)T cell	0.703 (0.678, 0.729)	0.630 (0.578, 0.682)
	Age + sex + all leukocyte subtypes	0.706 (0.680, 0.731)	0.643 (0.593, 0.693)
Stage III	Age + sex	0.653 (0.622, 0.683)	0.608 (0.568, 0.648)
	Age + sex + CD4(+)T cell	0.659 (0.629, 0.689)	0.622 (0.583, 0.660)
	Age + sex + all leukocyte subtypes	0.662 (0.632, 0.691)	0.626 (0.587, 0.665)
Stage IV	Age + sex	0.557 (0.520, 0.594)	0.557 (0.519, 0.595)
	Age + sex + CD4(+)T cell	0.597 (0.562, 0.631)	0.600 (0.564, 0.635)
	Age + sex + all leukocyte subtypes	0.593 (0.557, 0.629)	0.599 (0.562, 0.636)

## Data Availability

The data presented in this study are available on request from the corresponding author. The data are not publicly available due to ethical and data security requirements.

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
