# Peer review of "DNA Methylation-Based Estimates of Circulating Leukocyte Composition for Predicting Colorectal Cancer Survival: A Prospective Cohort Study"

_cancers, 2021, doi:10.3390/cancers13122948_

Round 1

Reviewer 1 Report

Considerable effort in data collection and in processing them have been done. Manuscript can be accepted after the revision of some points in  results section indicated below:

1-(lines 169-170, pg 4): the percentages of CRC patients with stage I,II, III and IV  were 18.1, 34.4, 32.9 and 14.0 respectively. In table 1 (pg.5) there are slightly different percentages (18.2,34.6,33.1,14.1). I suggest to uniform these.

2-(line 182 pg.5): check the note of the table 1. Remove "Numbers printed in bold.....". No comparisons are shown.

3-(line 194 pg.5): check HRs for the comparison of the highest with the lowest quartile of the neutrophils proportion;  text 1.32 (1.00, 1.27) vs figure 2 (1.00, 1.75).

Author Response

Considerable effort in data collection and in processing them have been done. Manuscript can be accepted after the revision of some points in  results section indicated below:

Response: Thank you very much for the very positive comments. The authors corrected all the typos mentioned in the following points and the corrections are highlighted.

Point 1-(lines 169-170, pg 4): the percentages of CRC patients with stage I,II, III and IV  were 18.1, 34.4, 32.9 and 14.0 respectively. In table 1 (pg.5) there are slightly different percentages (18.2,34.6,33.1,14.1). I suggest to uniform these.

Response 1: The authors have corrected the numbers in the text.

Point 2-(line 182 pg.5): check the note of table 1. Remove "Numbers printed in bold.....". No comparisons are shown.

Response 2: The authors have removed this redundant sentence.

Point 3-(line 194 pg.5): check HRs for the comparison of the highest with the lowest quartile of the neutrophils proportion;  text 1.32 (1.00, 1.27) vs figure 2 (1.00, 1.75).

Response 3: The authors have corrected the numbers in the text.

Reviewer 2 Report

The manuscript "DNA methylation-based estimates of circulating leukocyte composition for predicting colorectal cancer survival: A prospective cohort study" by Xin Gao et al uses leukocyte deconvolution of DNA methylation data of 2200 CRC patients to find associations of leukocyte composition with cancer survival.
The article is well presented. Background, aims, methods, results and statistics are comprehensive and seem sound. Discussion and conclusions are conclusive and realistic.

I expecially appreciate that the authors reflect on the strengths and limitations of their study in the discussion (lines 338 to 364), which appears authentic and honest.

In my opinion the article can be published in its current state.
However, the resolution of the following issues might be of importance to future readers: 

Material and Methods:
The EPIC methylation array method should be briefly outlined in 1 or 2 sentences.
e.g are the data from microarray or sequencing analysis? I might have overread it and after a quick google search I believe it is the former, but i'm still not sure.

Figure 3A:
CD4+ proportion in early stage seems to be associated with non-CRC-related death. Were the curves adjusted for CCI?

typos:
line 62: based
line 64: associated with

Author Response

The manuscript "DNA methylation-based estimates of circulating leukocyte composition for predicting colorectal cancer survival: A prospective cohort study" by Xin Gao et al uses leukocyte deconvolution of DNA methylation data of 2200 CRC patients to find associations of leukocyte composition with cancer survival.

The article is well presented. Background, aims, methods, results and statistics are comprehensive and seem sound. Discussion and conclusions are conclusive and realistic.

I expecially appreciate that the authors reflect on the strengths and limitations of their study in the discussion (lines 338 to 364), which appears authentic and honest.

In my opinion the article can be published in its current state. However, the resolution of the following issues might be of importance to future readers:

Response: Thank you very much for your appraisal of our manuscript! 

Point 1:

Material and Methods:

The EPIC methylation array method should be briefly outlined in 1 or 2 sentences.

e.g are the data from microarray or sequencing analysis? I might have overread it and after a quick google search I believe it is the former, but i'm still not sure.

Response 1: The genome-wide DNA methylation profile was analyzed using HumanMethylationEPIC BeadChip Kit (EPIC microarray). We have clarified it in lines 84-86 and lines 119-122.

Point 2:

Figure 3A:

CD4+ proportion in early stage seems to be associated with non-CRC-related death. Were the curves adjusted for CCI?

Response 2: Yes, among early-stage patients, the non-CRC deaths accounted for a large proportion of total deaths (85%). Because the survival rate of stage I CRC is high. The current survival curves in Figure 3 and Figure 4 were not adjusted for CCI. However, in the sensitivity analyses in supplementary materials, we observed similar results after additionally adjusting for CCI. The association of circulating CD4+ proportion and CRC-specific mortality was weaker than the association of circulating CD4+ proportion and all-cause mortality. It suggested that the DNA methylation-based leukocyte proportions were not specifically related to deaths from CRC, but general deaths. The authors discuss it in the manuscript. (Please see lines 337-341)

typos:

line 62: based

line 64: associated with

Response: The authors have corrected these typos.

Reviewer 3 Report

  1. This study reveals T lymphocytes are acritical for anti-tumor effect and overall survival. Because age was also associated to CD4+ T proportion (Table S1) and patients with advanced stage have received chemotherapy and/or radiotherapy (Fig S5), I feel it is inappropriate to confidently conclude that DNAm-based estimates of circulating leukocyte composition can predict CRC survival.
  2. Please divide tumor stage in Table 1 that we can see sex, age, and other factors in individual tumor stages. And please show the p value.
  3. In clinical practice, how to predict CRC survival using DNAm technique.

Author Response

Point 1: This study reveals T lymphocytes are acritical for anti-tumor effect and overall survival. Because age was also associated to CD4+ T proportion (Table S1) and patients with advanced stage have received chemotherapy and/or radiotherapy (Fig S5), I feel it is inappropriate to confidently conclude that DNAm-based estimates of circulating leukocyte composition can predict CRC survival.

Response 1: The authors appreciate the concerns from the reviewer. In this study, the statistically significant association of leukocyte composition with CRC prognosis remained even after controlling for age as well as other potential confounders such as therapies and comorbidities in the Cox regression models and survival curves. The C-index indicated that DNA methylation-based CD4+ cell proportion had a fairly good  predictive performance for CRC prognosis. Although it is still a long way to reach clinical practice, the proportion of CD4+ T cells have the potential to improve the accuracy of prognostic judgment for CRC patients. The findings of our study add to the existing body of knowledge in this field and provide a foundation for further study of inspiration.

Point 2: Please divide tumor stage in Table 1 that we can see sex, age, and other factors in individual tumor stages. And please show the p value.

Response 2: The authors have added a new supplementary table S1 as well as corresponding text in the manuscript according to the reviewer’s suggestion. (Please see lines 130-131 and lines 174-176)

Point 3: In clinical practice, how to predict CRC survival using DNAm technique.

Response 3: The authors acknowledge that one of the limitations of the DNAm technique is the high cost of the test. However, this study provides a potential biomarker for the prediction of CRC prognosis. Once a low-priced DNAm test is developed and commercialized, the DNAm-based leukocyte composition or other markers could assist the evaluation of patients’ prognosis as well as patients’ management in clinical settings. We add this part in the discussion section. (Please see lines 370-375)

Round 2

Reviewer 3 Report

Authors fully answer the questions. I suggest that the manuscript could be accepted for publication after 1 minor comment.

1. It may be Stage I to Stage IV in the group title in Table S1.

Author Response

Dear Reviewer, 

Thank you very much for the comment on the typos in Table S1. Please see the corrections in the supplementary materials.

Sincerely,

Xin Gao
